# Enhanced Detection of Desmoplasia by Targeted Delivery of Iron Oxide Nanoparticles to the Tumour-Specific Extracellular Matrix

**DOI:** 10.3390/pharmaceutics13101663

**Published:** 2021-10-12

**Authors:** Meenu Chopra, Jiansha Wu, Yen Ling Yeow, Louise Winteringham, Tristan D. Clemons, Martin Saunders, Venkata Ramana Kotamraju, Ruth Ganss, Kirk W. Feindel, Juliana Hamzah

**Affiliations:** 1Harry Perkins Institute of Medical Research, Centre for Medical Research, QEII Medical Centre, The University of Western Australia, Perth 6009, Australia; 21936806@student.uwa.edu.au (M.C.); susan.wu@perkins.org.au (J.W.); yen.yeow@perkins.org.au (Y.L.Y.); louise.winteringham@perkins.org.au (L.W.); ganss@perkins.org.au (R.G.); 2School of Molecular Sciences, The University of Western Australia, Perth 6009, Australia; Tristan.Clemons@usm.edu; 3School of Polymer Science and Engineering, University of Southern Mississippi, Hattiesburg, MS 39406, USA; 4Centre for Microscopy, Characterisation and Analysis, The University of Western Australia, Perth 6009, Australia; martin.saunders@uwa.edu.au (M.S.); Kwfeinde@dal.ca (K.W.F.); 5Cancer Research Center, Sanford Burnham Prebys Medical Discovery Institute, San Diego, CA 92037, USA; vraman@califiapharma.com; 6Curtin Health Innovation Research Institute, Curtin University, Perth 6102, Australia

**Keywords:** tumour targeting, extracellular matrix, CSG, nanoparticles, magnetic resonance imaging

## Abstract

Diagnostic imaging of aggressive cancer with a high stroma content may benefit from the use of imaging contrast agents targeted with peptides that have high binding affinity to the extracellular matrix (ECM). In this study, we report the use of superparamagnetic iron-oxide nanoparticles (IO-NP) conjugated to a nonapeptide, CSGRRSSKC (CSG), which specifically binds to the laminin-nidogen-1 complex in tumours. We show that CSG-IO-NP accumulate in tumours, predominantly in the tumour ECM, following intravenous injection into a murine model of pancreatic neuroendocrine tumour (PNET). In contrast, a control untargeted IO-NP consistently show poor tumour uptake, and IO-NP conjugated to a pentapeptide, CREKA that bind fibrin clots in blood vessels show restricted uptake in the angiogenic vessels of the tumours. CSG-IO-NP show three-fold higher intratumoral accumulation compared to CREKA-IO-NP. Magnetic resonance imaging (MRI) T_2_-weighted scans and T_2_ relaxation times indicate significant uptake of CSG-IO-NP irrespective of tumour size, whereas the uptake of CREKA-IO-NP is only consistent in small tumours of less than 3 mm in diameter. Larger tumours with significantly reduced tumour blood vessels show a lack of CREKA-IO-NP uptake. Our data suggest CSG-IO-NP are particularly useful for detecting stroma in early and advanced solid tumours.

## 1. Introduction

Clinical imaging techniques including magnetic resonance imaging (MRI), computed tomography (CT) and ultrasound can detect sporadically growing solid tumours, but their sensitivity and specificity decrease when detecting early-stage and metastatic diseases [1,2,3]. Moreover, the lack of sensitivity in both imaging-guided intraoperative visual inspection and manual palpation performed by the surgeon may result in incomplete resection of the tumour [4,5]. Hence, more sensitive and accurate detection strategies are needed for effective tumour removal and treatment. Furthermore, solid tumours typically display profound inter- and intratumoral molecular and phenotypic heterogeneity, not only in the cancerous cells, but also in the stromal cell and non-cell compartments [6,7]. Acquiring tumour biopsies prior to treatment is challenging as the procedure is invasive, takes longer to process and interpret and, consequently, is more expensive and not without risk to the patient [8]. Therefore, to improve the treatment outcome, it is important to improve diagnostic accuracy of tumour heterogeneity; molecular imaging is emerging as an important tool to achieve this. Importantly, this knowledge could inform the design of effective treatment strategies including patient stratification for better treatment options, monitoring therapeutic resistance and identifying metastatic disease.

Current systemic therapy and delivery of imaging agents to tumours rely mainly on the enhanced permeability and retention (EPR) effect of leaky tumour vasculature to deliver anti-cancer or imaging agents [9]. In diagnostic imaging, nanoparticle-based contrast agents, which have been traditionally exploited for cancer detection, have some limitations [10,11]. For example, iron oxide nanoparticles (IO-NP), including particles marketed as Feridex^®^ and Resovist^®^, failed commercially as contrast agents for solid tumours [12]. Their performance is poor partly because IO-NP do not penetrate sufficiently into solid tumours, and are taken up non-specifically in healthy surrounding tissue and rapidly cleared by the kidney or phagocytosed by the reticuloendothelial system in the liver and spleen [10,11,12].

To improve intratumoral delivery in solid tumours, imaging payloads including nanoparticle-based contrast agents can be actively targeted to bind to specific compartments in tumours using molecular ligands [13,14]. Widely used ligands include peptides that have high binding specificity to molecules aberrantly expressed in tumour blood vessels, hypoxic cells or immune cell infiltrates. Since the discovery of cell-adhesion molecules that are aberrantly expressed in tumours, which can be targeted by the RGD peptide [15,16], other tumour-targeting peptides have been discovered and used as effective ligands for systemic delivery of various drug and imaging payloads [17,18,19,20,21,22,23,24,25]. When directly coupled onto drugs or multivalently conjugated onto nanoparticles, tumour-targeting peptides can significantly enhance the intratumoral delivery of payloads, advancing their diagnostic and therapeutic use in multiple tumour types [23,26,27,28,29,30,31,32,33].

Solid tumours that are highly angiogenic can be effectively targeted with vessel homing peptides. For example, the RGD peptide specifically recognises α_v_β_3_ and α_v_β_5_ integrins on tumour endothelia [15], the NGR peptide selectively binds CD13 molecule in tumours but not in other CD13-expressing tissues [20] and the CREKA peptide binds to fibrin-clotted proteins commonly found in sprouting tumour blood vessels [23]. These vessel-targeting peptides have been extensively used to deliver radiotracers and nanoparticle-based contrast agents for multiple imaging modalities including MRI, ultrasound, CT and positron emission tomography (PET) [34,35,36,37,38]. However, the vessel network in solid tumours is highly heterogeneous in its distribution and function [39]. Tumours with fewer or less perfused blood vessels are harder to diagnose and they are likely to be less sensitive to vessel-targeted agents. Thus, alternative molecular targets for more effective delivery of imaging and therapeutic agents to enhance diagnostic and therapeutic strategies are needed.

The hallmarks of solid tumours associated with aggressive malignancies, poor prognosis and resistance to drug delivery include dense stroma with excessive production of ECM [40]. In treatment-resistant pancreatic adenocarcinoma and breast cancers, the fibrotic stroma and ECM may comprise 60–90% of the tumour mass [41,42,43]. The ECM in these tumours is made of highly abundant scaffolds of fibrillar proteins, glycoproteins and glycosaminoglycans, which are topographically and structurally different to the ECM of surrounding normal tissue [44]. The high ECM content can restrict tumour uptake of imaging agents and anti-cancer therapeutics [45]. In addition, desmoplastic cancers are more resistant to immunotherapy [46]; high ECM density hampers the migration of anti-tumour immune cells from reaching and interacting with the tumour cells [47] and downregulates the cytotoxic activity of anti-tumour T cells [48]. The abundant and aberrant tumour ECM has yet to be fully exploited by imaging technology as a means to diagnose cancer and evaluate novel therapies.

We have recently reported the discovery of a peptide, CSG (CSGRRSSKC) that binds specifically to the laminin-nidogen-1 complex of the ECM in mouse and human breast, pancreas and liver tumours [49]. We have shown that CSG fused to a cytokine, TNFα, acts as a delivery agent allowing for specific binding of TNFα to tumour ECM, triggering effective immune-mediated anti-tumour effects [49].

In this study, we use CSG to deliver IO-NP to the ECM compartment of solid tumours. Using a murine PNET model that spontaneously develops angiogenic tumour nodules with significant ECM content, we compare the intratumoral uptake of CSG-targeted IO-NP to control untargeted and CREKA-targeted IO-NP.

## 2. Materials and Methods

### 2.1. Peptide

CSG is a 9-amino acid cyclic peptide (CSGRRSKC), referred to as CSG, based on its first three amino acids, and CREKA is a linear pentapeptide (Cys-Arg-Glu-Lys-Ala). Briefly, peptides, amidated at the C-terminus, were synthesised on a microwave-assisted automated peptide synthesiser (Liberty; CEM, Matthews, NC, USA) following Fmoc/tertiary butyl strategy on rink amide MBHA resin with HBTU (*N*,*N*,*N*′,*N*′-tetramethyl-*O*-(1H-benzotriazol-1-yl) uronium hexafluorophosphate) activator (or *O*-(benzotriazol-1-yl)-*N*,*N*,*N*′,*N*′-tetramethyluronium hexafluorophosphate), collidine activator base and 5% piperazine for deprotection. Fluorescein was incorporated during the synthesis at the N-terminus of the sequence as 5(6)-carboxyfluorescein, with a spacer, 6-aminohexanoic acid, separating the fluorophore and the sequence. All the peptides were purified to purities > 90% by HPLC with acetonitrile and water gradient mixtures with 0.1% TFA. Peptides were analysed by MALDI and found to have [M+H] 1554.75 for FAM-cys-CSG, and 1076.24 for FAM-CREKA.

### 2.2. PNET Mouse Model

RIP1-Tag5 mice (on a C3H background, provided by D. Hanahan) express the oncogene SV40 Large T antigen (Tag) under the control of the rat insulin gene promoter (RIP) in pancreatic β cells, and develop spontaneous insulinoma as previously described [50]. The experimental protocol was approved by the Animal Ethics Committee of the University of Western Australia (UWA). RIP1-Tag5 mice at ≈29–30 weeks of age were used in the studies, as they are known to develop advanced-stage tumour nodules. The mice were kept under pathogen-free conditions. A minimum of *n* = 3 mice per group were used in each study.

### 2.3. In Vivo Peptide Binding

Tumour-bearing mice were injected intravenously (tail vein) with 100 µL of either 1 mM FAM-CSG or FAM-CREKA in phosphate saline buffer (PBS). After 1 h, animals were euthanised and perfused by transcardial perfusion with sterile PBS to remove the unbound peptide. Tumours were excised, embedded in O.C.T. (Tissue-Tek^®^) as frozen (unfixed) and stored at 80 °C for histology analysis.

### 2.4. Synthesis of Targeted Iron Oxide Nanoparticles

Superparamagnetic iron oxide nanoparticles (IO-NP) were prepared as per previously established protocols [51] with slight modification. Briefly, the homogenous IO-NP was made by a co-precipitation method in an aqueous solution with a molar ratio of Fe (II) to Fe (III) = 0.5 and pH ≈ 11–12. In brief, 5.2 g of FeCl_3_ and 2.0 g of FeCl_2_ were dissolved in the aqueous solution and added dropwise into 250 mL of 1.5 M NaOH solution under vigorous stirring. The resulting solution was heated at 70 °C for 2 h. A black precipitate was generated, purified deoxygenated water was added and the solution was centrifuged at 4000 rpm. The resulting nanoparticles were anionic, the anionic charge of the nanoparticles was neutralised by adding 500 mL of 0.01M HCI solution to the precipitate (with stirring) [51]. The colloidal nanoparticles were again separated by centrifugation (4000 rpm). The bare IO-NP were then non-covalently coated with a thin layer of dextran (MW 5220, Sigma, Australia) as previously described [52]. Fluorescein-peptide-lipid labelling was prepared by coupling 1,2-distearoyl-sn-glycerol-3-phosphoethanolamine-*N*-maleimide (DSPE-PEG2000-maleimide, Avanti Polar Lipids) with peptide carboxyfluorescein (FAM) bearing a cysteine on its *N*-terminus in 1:1 molar ratio. The dextran-coated IO-NP were encapsulated in PEGylated lipids by combining the dextran-coated IO-NP with DSPE-PEG2000 and DSPE-PEG-maleimide-2000-FAM-peptide (in chloroform) in 1:1:0.37 molar ratios [24,53]. The iron concentration was determined by a standard curve prepared from the stock solution (FeSO_4_, 40 mg/L). The stock solution at different concentrations (0.5–5 mg/mL) was dissolved in acid (0.2% hydroxylamine hydrochloride) and sodium citrate was added to maintain an acidic pH for Fe, forming the phenanthroline/Fe_2_ complex. The quantification of the known dilution was done spectrophotometrically by adding o-Phenanthroline (0.0075% *w*/*v*) into each solution [54].

### 2.5. Material Characterisation

Particle size, morphology, crystallinity and size distribution were obtained using high-resolution transmission electron microscopy (HR-TEM) and selected area electron diffraction (SAED) carried out on a FEI Titan G2 80-200 TEM at an accelerating voltage of 200 kV. To determine bare IO-NP size and morphology, a drop of IO-NP solution was placed on the surface of a carbon-coated copper grid and dried at room temperature overnight before imaging using HR-TEM. The average hydrodynamic size and zeta potential of all IO-NP were characterised using dynamic light scattering (DLS, Nano ZS90; Malvern Instruments, Malvern, UK), using a 4 mW He-Ne laser operating at 633 nm with a scattering angle of 173°. Measurements were taken in triplicate after an initial equilibration period of 2 min. For calibration, the measurement’s ‘material’ was defined as nanoparticles (refractive index of 1.515 and absorbance of 0.05) and ‘dispersant’ was defined as water at 25 °C (refractive index of 1.330 and viscosity of 0.887). The volume-weighted hydrodynamic radius and zeta potential of CSG-IO-NP, CREKA-IO-NP and untargeted-IO-NP were presented as mean ± standard deviation.

### 2.6. In Vitro IO-NP MRI

MRI scans were performed at 9.4 T with a Bruker BioSpec 94/30 magnet, Avance III HD console and ParaVision 6.0.1 acquisition software. To determine the transverse relaxation time (T2) of IO-NP, IO-NP (after encapsulation in PEG) were prepared at different concentrations of Fe (0.02, 0.05, 0.07, 0.18 and 0.36 mM of Fe) in PBS. The T2 data were collected using a 2D multi-slice multi-echo (MSME) sequence with the following parameters: echo time (TE) = 1.5 ms, repetition time (TR) = 2000 ms, slice thickness = 0.5 mm, pixel size = 100 µm × 100 µm, number of echoes = 64, 4 averages and total experiment time = 16 min. Bruker ParaVision and ImageJ software were used for MR image reconstruction and analysis.

### 2.7. Ex Vivo Tissue MRI

A 100 μL bolus of CSG-IO-NP, CREKA-IO-NP and untargeted IO-NP (5 mg/kg Fe) were injected intravenously (tail vein) into the RIP1-Tag5 mice. After 4 h circulation, mice were perfused with sterile PBS; then, tissues were collected and fixed in 2% formalin for 2 h and embedded in 2% agarose. Following the MRI localizer scan, a series of three scans were completed with 23 interleaved coronal slices: (1) a T2*-weighted image (FLASH: echo time (TE) = 10 ms; repetition time (TR) = 600 ms; flip angle (FA) = 30°; averages (NA) = 2; acquisition time (TA) = 3 min); (2) T2* map (multi-gradient echo (MGE): TR = 1250 ms; TE = 2.6 ms; FA = 90°; number of echoes (NE) = 10; echo spacing (ESP) = 5 ms; TE (effective) = 2.6 to 47.6 ms; and TA = 5 min); and (3) a T2 map (MSME: TR = 3150 ms; TE = 8 ms; FA = 90°; refocusing pulse = 180°; NE = 16; ESP = 8 ms; TA = 7 min 53 s). For the above scans, the field of view (FOV) = 3.6 × 3.6 cm^2^, matrix (MTX) = 240 × 240, slice thickness (ST) = 0.5 mm and slice gap = 25 µm. T2* and T2 parameter maps were calculated from the MGE and MSME datasets, respectively, using the ParaVision 6.0.1 macro *fitinlsa*, which fits the signal for each pixel according to a mono-exponential decay. Image analysis was performed using ImageJ. Tumour volume and IO-NP-induced changes in MRI signal level were measured by using the *Image Display and Processing Tool*. ROIs for each tumour was manually defined using the *track* tool, based on the MSME image with TE = 8 ms. Statistics for tumour volumes were obtained by combining the ROI statistics on an image slice by slice basis.

### 2.8. In Vivo Lectin Perfusion

For lectin perfusion, the mice were i.v. injected with 100 μg of FITC-labelled tomato lectin (Lycopersicon esculentum; Vector). After 10 min of circulation, mice were heart-perfused with 2% neutral-buffered formalin (*w*/*v*) and tumours were frozen in the O.C.T. compound.

### 2.9. Immunofluorescence and Immunohistochemistry Analysis

Tumours collected for peptide and IO-NP binding and post-MRI were analysed. Tissue distribution of fluorescein (FAM)-labelled CSG, FAM-CREKA, FAM-IO-NP, FAM-CSG-IO-NP or FAM-CREKA-IO-NP were detected on 8 μm tissue cross-sections based on their fluorescence intensity and reactivity to anti-fluorescein-HRP antibody (polyclonal, GeneTex) and counter-stained with haematoxylin or methyl green (Vector Laboratories, Burlingame, CA, USA). For co-staining analysis, the following antibodies were used: anti-CD31 (390; ebioscience), anti-laminin (polyclonal; Millipore) and anti-collagen I (polyclonal; Abcam). For secondary detection, fluorescence-labelled, 594-conjugated anti-rat, rabbit or goat IgG (Life Technologies, Carlsbad, CA, USA) were used. Images were captured on a Nikon Ti-E microscope (Nikon Instrument Inc., Melville, NY, USA) or ScanScope XT (Aperio Technology, Inc., Vista, CA, USA). Image analysis and quantification were performed either using NIS software modules (version 4.0) or ImageScope version 12.1.0.5029 (Aperio Technology, Inc., Vista, CA, USA).

### 2.10. Statistical Analysis

A minimum of 3 mice per group was used in all studies. For histological quantification, raw data were obtained from each microscopically identified tumour on tissue section (1–3 identified tumours/tissue sections). Raw data were obtained from at least three fields of view for tumours > 3 mm or from the entire tumour section for tumours < 3 mm. Statistical analyses were performed using GraphPad Prism 7 (GraphPad Prism Software, San Diego, CA, USA). Data were analysed by the Student’s *t*-test (two-tailed) or one-way analysis of variance (ANOVA). A *p* value < 0.05 was considered statistically significant. Error bars indicate SEM. Experiments were carried out in an unblinded fashion.

## 3. Results

### 3.1. CSG and CREKA Have Distinct Binding Targets in PNET In Vivo

We first compare the in vivo CSG and CREKA binding to RIP1-Tag5 tumours following an i.v. injection of each FAM-labelled peptide. Figure 1 compares the peptide intratumoral distribution, which matches our previous in vitro binding assessment [49]. Following i.v. injection, both peptides accumulate intratumorally. CREKA accumulation is specifically confined to the CD31-positive tumour blood vessels (Figure 1A). In contrast, CSG accumulates mostly in areas that are outside of CD31-positive tumour blood vessels (Figure 1B), where it colocalises with tumour laminin (Figure 1C). These in vivo binding data confirm different binding targets for CSG and CREKA in PNET, which we seek to exploit for targeted delivery of IO-NP. Previous studies have shown that CSG and CREKA have limited binding in normal healthy tissues [23,49].

### 3.2. Synthesis and Characterization of IO-NP

The IO-NP synthesised in this study were first characterised in terms of size, shape and composition (Figure 2). Structural characterisation of bare IO-NP was performed using high-resolution transmission electron microscopy (HR-TEM) and selected area electron diffraction (SAED). The size and morphology of bare IO-NP, as shown in Figure 2A, are uniformly spherical and ≈10 nm in diameter. The lattice fringe spacings evident from HR-TEM are measured to be ≈2.96 Å, corresponding to the (220) plane of magnetite (Fe_3_O_4_) [55]. To further confirm the IO-NPs were of the magnetite phase, the SAED pattern was obtained from the IO-NP sample (Figure 2B). Based on this diffraction pattern, the measurement of the lattice spacings was consistent with the atomic spacings anticipated for magnetite, Fe_3_O_4__ IO-NPs (Table 1) [55].

As illustrated in Figure 3, our imaging agents consist of three main elements—(i) dextran-coated IO, encapsulated by (ii) PEGylated lipids and (iii) PEGylated lipids—which were tagged with either FAM-labelling (untargeted), FAM-CREKA or FAM-CSG via cysteine-maleimide conjugation. The final hydrodynamic size of the untargeted IO-NP, CREKA-IO-NP and CSG-IO-NP are between 28 and 35 nm, and all IO-NP have a positive surface charge after conjugation, as measured by DLS (Table 2). The indicated morphological characteristics, superparamagnetic core, hydrodynamic sizes of less than 200 nm and positive overall surface charges of our synthesised IO-NPs are comparable to the previously reported MRI contrast agents suitable for biomedical and in vivo applications [56,57].

### 3.3. Enhanced Intratumoral ECM Binding of CSG-IO-NP Compared to Untargeted IO-NP and CREKA-IO-NP

We compare the intratumoral distribution and levels of CSG-IO-NP to the untargeted-IO-NP and CREKA-IO-NP, following 4 h of in vivo circulation (Figure 4). Particle distribution was confirmed by histological detection using an antibody against the fluorescein label (Figure 4). The untargeted IO-NP show limited intratumoral tumour accumulation. Both CREKA-IO-NP and CSG-IO-NP show specific intratumoral accumulation and minimal binding to exocrine tissue surrounding the tumours (Figure 4A). The accumulation of CSG-IO-NP in tumours is at least three-fold higher than in the CREKA-IO-NP tumour binding. (Figure 4B). CREKA-IO-NP binding in tumours correlates with the location of CD31-positive endothelia (Figure 5A), whereas CSG-IO-NP binding correlates mainly with tumour ECM detected with collagen I, laminin and nidogen-1 staining (Figure 5B). The finding is consistent with the intratumoral distribution of CREKA and CSG as free peptides (Figure 1), suggesting that both CSG and CREKA specifically deliver IO-NP to their respective target.

### 3.4. CSG-IO-NP Is an Effective MRI Contrast Agent, Irrespective of Tumour Sizes

To assess the use of IO-NP as an effective MRI contrast agent, we first evaluated T2*-weighted and T2 relaxation MR images for a range of concentration of IO-NP from 0.02 to 0.36 mM Fe in vitro. Figure 6A demonstrates the effect of different iron concentrations on T2, consistent with previous reports [53,56].

We then performed ex vivo MRI scans of RIP1-Tag5 tumours with exocrine pancreatic tissue following the 4 h in vivo circulation of CSG-IO-NP, CREKA-IO-NP and untargeted IO-NP in tumour-bearing mice. Figure 6C–E show T2* and T2 relaxation images of scanned tumours based on their small (diameter < 3 mm) and large (diameter > 3 mm) sizes. Based on changes in the T2 relaxation time that indicate the relative amount of IONP in tumours, our data (Figure 6F) show significant CSG-IO-NP intratumoral accumulation irrespective of tumour size. CREKA-IO-NP are consistently detectable only in small tumours, and the untargeted IO-NP show limited uptake in all tumours.

Since tumour sizes affect only CREKA-IO-NP intratumoral binding, we compared the blood vessel and ECM composition in small and larger sized tumours. The RIP1-Tag5 tumours are known for their blood vessel heterogeneity with advanced insulinoma showing poor perfusion [50,58]. To identify ‘functional’ blood vessels, we compared lectin-painted tumour blood vessels (following i.v. injection of FITC-labelled lectin, a surrogate marker for tumour perfusion) [30,59] with collagen-I-positive ECM content (Figure 7A). Our data (Figure 7B) show lectin-positive vessels are significantly reduced in larger tumours (diameter > 3 mm), whereas there are no changes in the ECM content in all tumours irrespective of tumour sizes. This finding suggests that CSG-IO-NP may be useful for a broader range of tumour sizes due to their ECM abundance.

## 4. Discussion

Using the CSG peptide to target aberrant ECM in solid tumours, we developed a targeted IO-NP for effective intratumoral delivery. Our CSG-IO-NP are particularly useful as an MRI contrast agent to mark tumours based on their content of desmoplastic stroma. In PNET tumours that are intratumorally resistant to the untargeted IO-NP, we found CSG-IO-NP substantially overcome the EPR limitation. Their intratumoral accumulation and binding to ECM, at least in the RIP1-Tag5 tumours, are not limited by changes in tumour size or tumour blood vessel content.

Solid tumours often develop resistance to systemic delivery [45,60]. Hence, nanoparticle-based imaging agents have been commonly used to detect tumour nodules based on contrast enhancement generated by their accumulation in the normal and marginal tissues and lack of uptake in tumours. For example, in a liver cancer model, surface-enhanced Raman scattering (SERS) nanoparticles were used based on their efficient homing to healthy liver tissue but not to intrahepatic malignancies [61]. Similarly, the pre-clinical and clinical use of MRI- and PET-based contrast agents is often based on their contrast enhancement that allows for the delineation of small peritumoral vessels surrounding the tumour and tumour rim [60,61,62]. In this study, we used the RIP1-Tag5 tumours for imaging assessment because these tumours, while highly angiogenic, are poorly perfused and largely impenetrable to drugs and imaging agents [49,58]. The imaging contrast generated by MRI scans shows traces of untargeted IO-NP only around the marginal tissue and not in the tumour parenchyma (Figure 6), confirming the intratumoral barrier that forms in solid tumours. These data support our previous findings [49] and imaging studies in other experimental and clinical tumours [61,63,64].

An alternative approach for contrast enhancement is to increase the uptake of the imaging agent in tumours and reduce the non-specific binding in the surrounding tissue. This is achieved through the use of imaging agents that effectively bind or are reactive to the expression of specific molecular characteristics that are unique to tumours and are not present in the surrounding healthy tissues [13,14]. This approach relies on unique specificity and abundance of the molecular targets [13,14]. In this study, we increase the intratumoral uptake of IO-NP using the tissue homing peptides CREKA and CSG. We show CREKA and CSG, when tagged onto IO-NP, retain their binding to tumour blood vessel and ECM targets, respectively.

Our histology analysis and MRI scans of tumours following the i.v. injection of CREKA-IO-NP and CSG-IO-NP provide functional and molecular information on the state of the tumour microenvironment at early and advanced stages. CREKA binding to the fibrin-clotted protein that localises within angiogenic vessels has been widely used to target drugs and nanoparticles to tumour blood vessels for theragnostic purposes [23,27,31,32,36,38]. Similarly, in RIP1-Tag5 tumours, CREKA-IO-NP localise within blood vessels in relation to vessel numbers and functionality. Reduced intratumoral uptake of CREKA-IO-NP correlates with a lower number of lectin-painted blood vessels in larger RIP1-Tag5 tumours, indicating that poorly perfused blood vessels in larger size tumours are less receptive to CREKA targeting. This finding suggests heterogeneity of blood vessel functionality at different stages may limit the use of vessel-targeted imaging agents.

Previously, we showed that the target for CSG binding, the laminin-nidogen-1 complex, is an extensive network formed throughout the RIP1-Tag5 tumours and overlaps with fibrillar collagens including collagen-1 [49]. RIP1-Tag5 tumours express two- to three-fold higher laminin-nidogen-1 complex than the normal exocrine pancreas. Importantly, unlike normal tissue ECM that is expressed exclusively as a thin basement membrane supporting vessels and epithelia, in tumours, laminin and nidogen-1 are also aberrantly expressed as avascular ECM [49]. These findings raise the possibility that structural differences in tumour ECM, absent in normal ECM, expose an otherwise hidden epitope for CSG binding. CSG binding is highly specific for tumour ECM and not ECM in normal tissue [49]. Analysis of IO-NP distribution in RIP1-Tag5 tumours shows CSG-IO-NP are dispersed within tumour ECM, well away from the nearest angiogenic vessels, with the overall intratumoral amount of CSG-IO-NP at least double that of CREKA-IO-NP. Here, we show the abundance of ECM in RIP1-Tag5 tumours is consistent in small and large tumours, resulting in a reliable intratumoral accumulation of CSG-IO-NP, unlike CREKA-IO-NP, which are dependent on tumour size and blood vessel content.

CSG-IO-NP have two important potential applications. Firstly, CSG-IO-NP may have potential as an imaging agent for a wide range of solid tumours beyond PNET. The laminin-nidogen-1 complex is highly conserved in mouse and human tumours and clearly represents a robust molecular target. We have previously shown that CSG specifically recognises ECM in murine transplant and/or transgenic breast, liver and colon cancers, using i.v. injection or by direct in vitro binding. CSG binding is also conserved in human pancreatic adenocarcinoma, as well as breast and hepatocellular carcinoma [49], highlighting its potential clinical use.

Secondly, we developed a new biologic agent, TNFα-CSG, which is an immune-modulating cytokine that binds specifically to tumour ECM. TNFα-CSG treatment in preclinical cancer models effectively promotes intratumoral accumulation of cytotoxic T cells and suppresses tumour growth without systemic toxicity or increase in metastatic activity [49]. CSG targeting of TNFα is critical to induce vessel dilation and improve tumour perfusion [49]. This outcome is different from the anti-vascular effect of TNFα, where TNFα-targeted peptides bind to tumour blood vessels or ECM proteins that are primarily located around tumour blood vessels [29,65,66]. Whilst we are interested in developing TNFα-CSG as an anti-cancer therapeutic, as an imaging agent, CSG-IO-NP may be used for diagnostic screening to identify patients who may benefit from TNFα-CSG therapy.

CSG targeting of TNFα and IO-NP offers complementary therapeutic and diagnostic agents for fibrotic cancers that have aberrant laminin-nidogen-1 expression. The use of CSG-IO-NP as a molecular imaging agent for detecting hypovascularized and fibrotic tumours such as pancreatic ductal adenocarcinoma remains to be investigated.

## Figures and Tables

**Figure 1 pharmaceutics-13-01663-f001:**
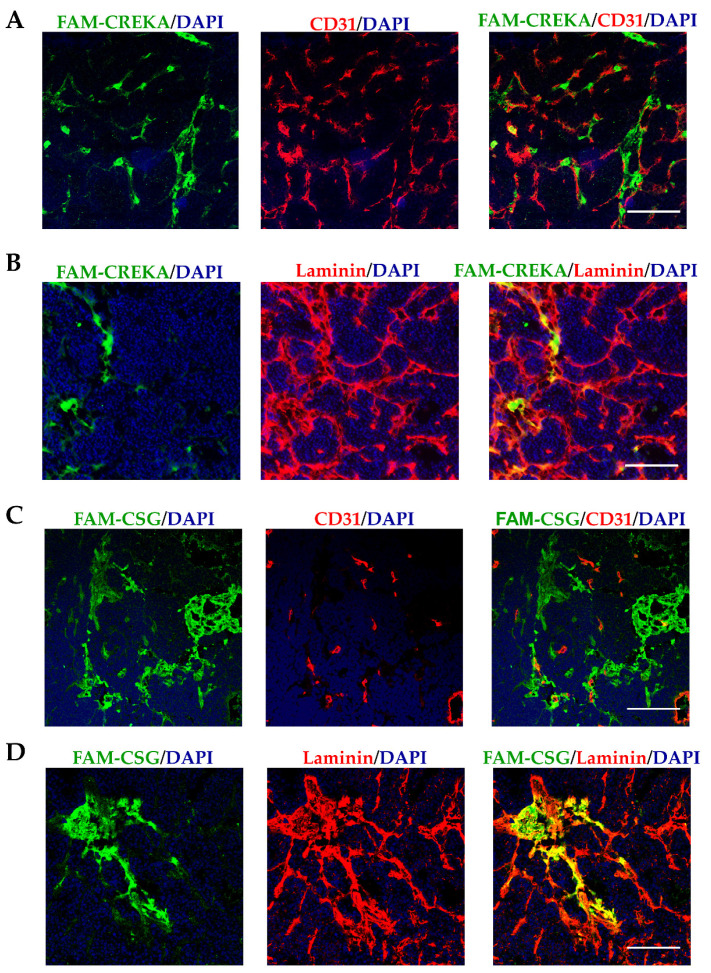
CSG and CREKA specifically recognise different compartments in RIP1-Tag5 tumours. Tumour-bearing RIP1-Tag5 mice were i.v. injected with 0.1 µmol of FAM-CREKA or FAM-CSG, and tissues were collected after 1 h of circulation. (**A**–**D**) show distribution of FAM-CREKA and FAM-CSG (green) in tumours co-stained either with the vascular marker CD31 or the ECM marker laminin (red). Scale bars: 50 μm.

**Figure 2 pharmaceutics-13-01663-f002:**
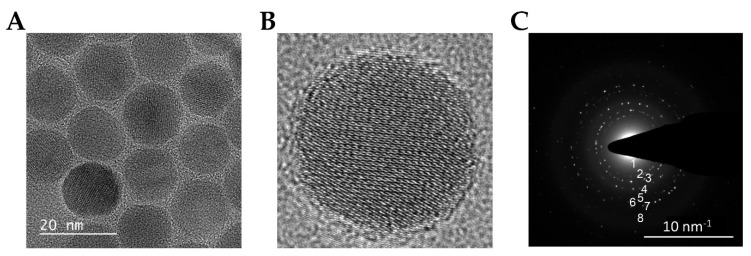
Structural and morphological characterisation of IO-NPs. (**A**) High-resolution transmission electron microscopy (HR-TEM) micrographs of IO-NPs and (**B**) a magnified view of a single NP. (**C**) Selected area electron diffraction (SAED) pattern acquired from an IO-NP assembly.

**Figure 3 pharmaceutics-13-01663-f003:**
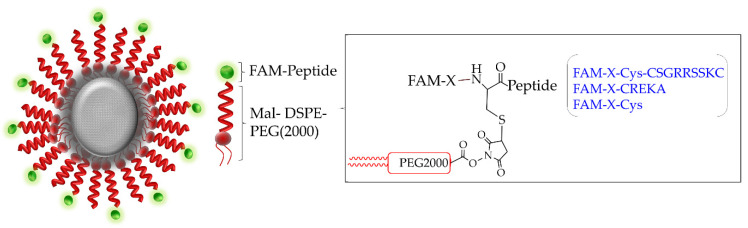
Schematic illustration of IO-NP consisting of a dextran-coated IO particle (core, gray) encapsulated by DSPE-PEG2000 lipids (in red) with the lipids tagged to a peptide or cysteine residue (in green). FAM-X- Cys-CSG (CSGRRSSKC), FAM-X-CREKA or FAM-X-Cys (untargeted) is tagged to the lipids via cysteine-maleimide (Mal) interaction (shown in the box).

**Figure 4 pharmaceutics-13-01663-f004:**
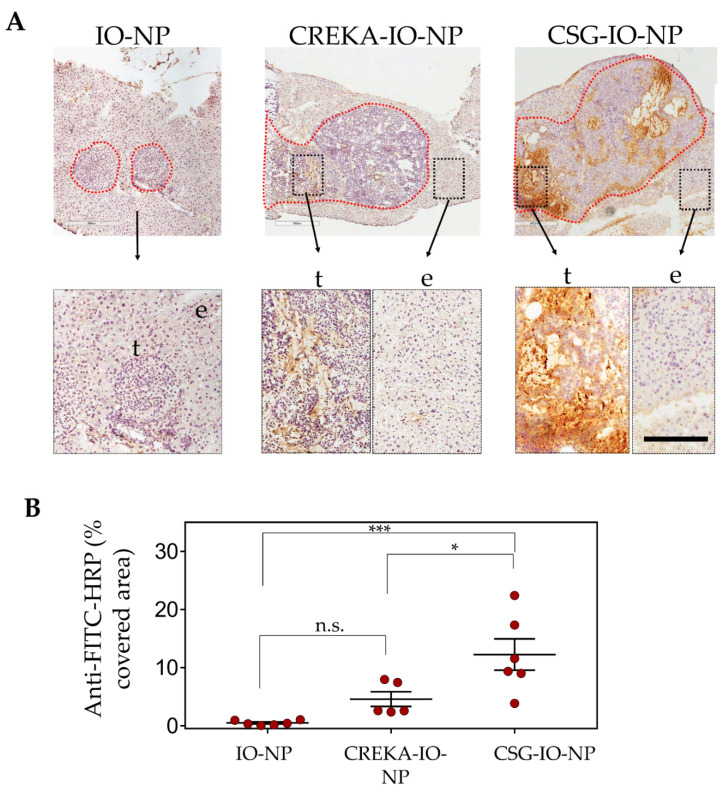
CSG and CREKA deliver IO-NP to a specific target in solid tumours. Mice bearing RIP1-Tag5 tumours were i.v. injected with 100 μL of the indicated IO-NP (5 mg/kg Fe), and tissues were collected after 4 h of circulation. (**A**) Top: Representative micrographs of tumours (circled, red) with surrounding exocrine pancreas. Nanoparticle uptake in these tissues was detected based on anti-FITC-HRP reactivity (brown) and nuclei were counterstained with haematoxylin. Scale bar, 200 µm. Bottom: Higher magnification of the tumour area (t) and exocrine pancreas (e). (**B**) Plots of % area per tumour that was positive for anti-FITC antibody detection (in **A**) for tumours with untargeted-IO-NP, CREKA-IO-NP and CSG-IO-NP, and mean ± SEM (*n* = 3 mice/group *** *p* < 0.005, * *p* < 0.05, n.s. = not significant, by one-way ANOVA test).

**Figure 5 pharmaceutics-13-01663-f005:**
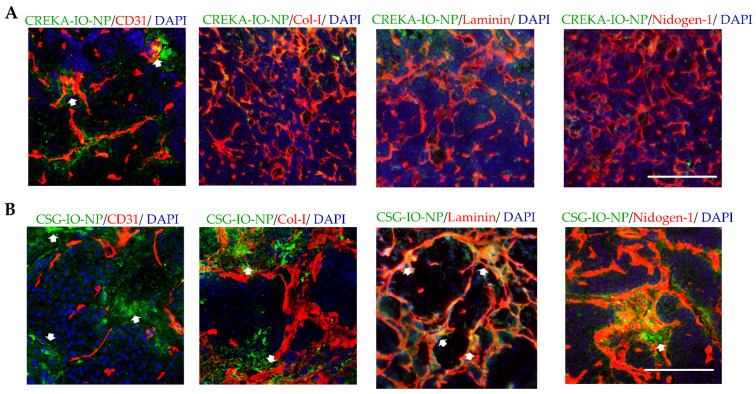
RIP1-Tag5 tumours with (**A**) CREKA-IO-NP and (**B**) CSG-IO-NP (green) were co-stained with CD31 (red), collagen I (Col-1, red), laminin (red) or nidogen-1 (red), and nuclei were stained with DAPI (blue). Representative micrographs are shown with indicated detection of IO-NP accumulation (arrows). Scale bars, 50 µm.

**Figure 6 pharmaceutics-13-01663-f006:**
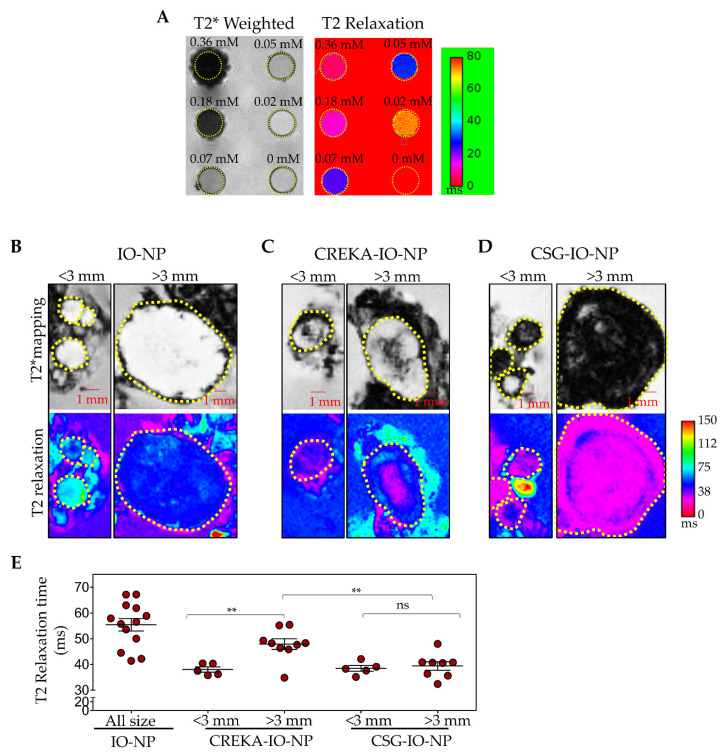
CSG-IO-NP show greater MRI tumour contrast compared to untargeted or CREKA-IO-NP. (**A**) T2*-weighted images (black and white) and T2 relaxation maps (coloured) of IO-NP at indicated concentrations ranging from 0 to 0.36 mM Fe. (**B**–**D**) Representative images of T2*-weighted (top) and T2 relaxation maps (magenta, bottom) of tumours with exocrine pancreas following ex vivo MRI scan. Tumours were analysed based on their sizes (<3 mm or >3mm in diameter). Scale bar: 1 mm. RIP1-Tag5 tumours and exocrine tissue were isolated after 4 h of in vivo circulation of untargeted-IO-NP, CREKA-IO-NP and CSG-IO-NP in tumour-bearing mice (*n* = 10–15 tumours from *n* = 5–6 mice per group, 5 mg/kg Fe). Mice were perfused with PBS to remove the unbound particles. The MRI scan was performed on formalin-fixed tumours embedded in 2% agarose. (**E**) Reduction in T2 relaxation time indicates the relative increase of IONP in individual tumours and mean ± SEM (** *p* < 0.01 by Student’s *t*-test. n.s. = not significant).

**Figure 7 pharmaceutics-13-01663-f007:**
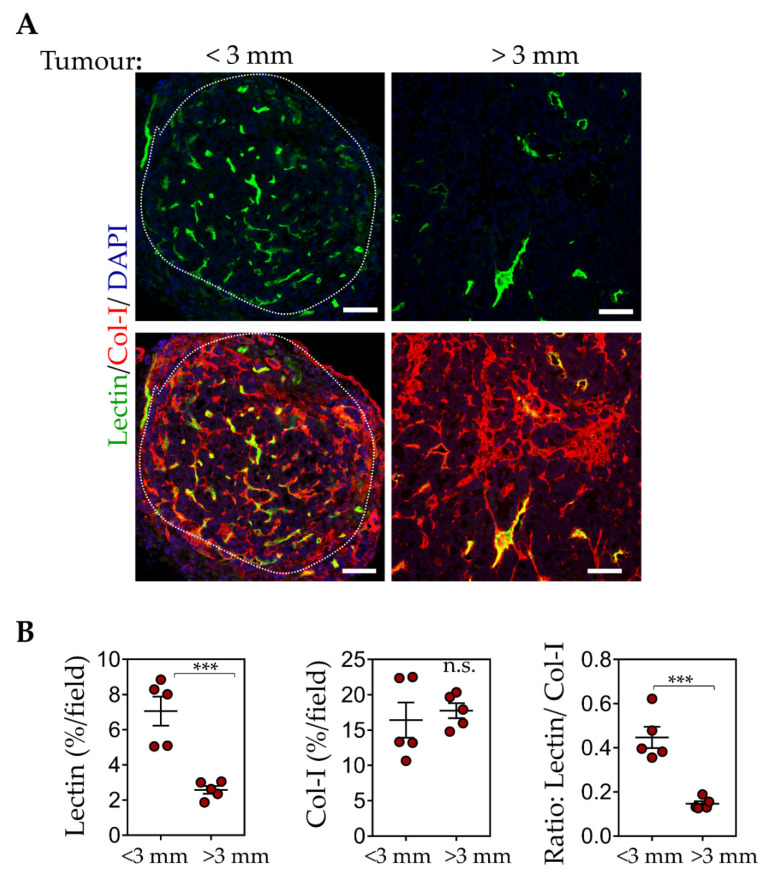
ECM and blood vessel contents in tumours correlate with the effectiveness of CSG and CREKA to deliver IO-NP. (**A**) Lectin^+^ vessels (green) in tumours after RIP1-Tag5 mice received an i.v. injection of FITC-labelled lectin (0.1 μmol). Top: Representative micrographs show % of lectin^+^ vessels in small tumours (diameter < 3 mm) and larger tumours (diameter > 3 mm). Bottom: The same micrographs indicating lectin+ area and collagen I expression (red). Scale bars: 100 μm. (**B**) Quantification of % lectin^+^/tumour (left), % collagen-I+ expression/tumour (middle) and ratio of lectin:collagen-I (right) with mean ± SEM are shown (10 tumours from *n* = 3 RIP1-Tag5 mice (*** *p* < 0.001 for Student’s *t*-test).

**Table 1 pharmaceutics-13-01663-t001:** Measured lattice spacing, d (angstroms), based on the rings from the selected area electron diffraction (SAED) pattern in Figure 2C, along with their respective hkl miller indices.

Ring	1	2	3	4	5	6	7	8
d (Å)	4.88	2.96	2.52	2.08	1.68	1.62	1.47	1.26
hkl	111	220	311	400	422	511	440	533

**Table 2 pharmaceutics-13-01663-t002:** IO-NP sizes and their zeta potentials as measured by dynamic light scattering (DLS).

NPs	Size (nm)	Zeta Potentials (mv)
Untargeted FAM-IO NP	28.31 ± 4.77	32.20 ± 2.26
FAM-CREKA-IO-NP	35.40 ± 8.10	38.43 ± 7.10
FAM-CSG-IO-NP	33.83 ± 5.57	34.88 ± 4.83

## Data Availability

Data are available upon request.

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
