# Peer review of "Enhanced Detection of Desmoplasia by Targeted Delivery of Iron Oxide Nanoparticles to the Tumour-Specific Extracellular Matrix"

_pharmaceutics, 2021, doi:10.3390/pharmaceutics13101663_

Round 1

Reviewer 1 Report

The authors demonstrate in this study that the use of superparamagnetic iron-oxide nanoparticles (IO-NP) conjugated to a tumor targeting peptide called CSG accumulate specifically in tumors. The experiments have been carried in careful fashion and the methodology for tumor targeting studies is impressive. The study is further enhanced by comparing CSG-conjugated nanoparticles not only to unconjugated nanoparticles (no peptide attached), but also to the nanoparticles coupled to another tumor targeting peptide (representing additional control/competition). This tumor targeting peptide (CREKA) has itself gained (according to the review of the literature) a reputation as a potentially useful targeting device in tumor diagnostics and drug delivery. I have following comments/criticism for the manuscript:

  1. 1. Double-staining of control peptide and laminin needs to be presented for comparison.
  2. 4C. Double-staining of control peptide and collagen type I needs to be presented for comparison.
  3. 4. Tumors could be stained for laminin/nidogen/fibrin to demonstrate whether the homing pattern of the peptides is identical to that of their target receptors in the tumors.
  4. I do not understand the mechanism how the CSG peptide targets tumors in relation to its molecular target. Namely, if its receptor is laminin-nidogen complex, the result would imply that the peptide accumulates in all tissues, but it is retained in tumor tissue due to binding to its receptor in the ECM. Please expand the discussion and address whether CSG peptide has a receptor expressed selectively in tumor blood vessels. By having “tissue”/disease selective receptors expressed by the angiogenic blood vessels in tumors, one could explain the selective accumulation of CSG peptide in the ECM of the tumor tissue. If the receptor is indeed only in the ECM, the accumulation of the peptide could be due to enhanced retainment of the peptide in the tumor matrix. Please address this by explaining the tumor targeting by the CSG peptide.

Author Response

Dear Reviewer,

Thank you very much for your supportive feedback. We have now addressed your comments/suggestions.

  1. Double-staining of control peptide and laminin needs to be presented for comparison.

ANSWER: We have now included double staining of CREKA and laminin in Figure 1.

4C. Double-staining of control peptide and collagen type I needs to be presented for comparison

ANSWER: We have now included co-staining of CREKA-IO-NP and collagen I, laminin and nidogen-1, as a new Figure 5A.

  1. Tumors could be stained for laminin/nidogen/fibrin to demonstrate whether the homing pattern of the peptides is identical to that of their target receptors in the tumors.

ANSWER: We have now included co-staining of CSG-IO-NP and collagen I, laminin and nidogen-1, as a new Figure 5B.

I do not understand the mechanism how the CSG peptide targets tumors in relation to its molecular target. Namely, if its receptor is laminin-nidogen complex, the result would imply that the peptide accumulates in all tissues, but it is retained in tumor tissue due to binding to its receptor in the ECM.

Please expand the discussion and address whether CSG peptide has a receptor expressed selectively in tumor blood vessels.

By having “tissue”/disease selective receptors expressed by the angiogenic blood vessels in tumors, one could explain the selective accumulation of CSG peptide in the ECM of the tumor tissue. If the receptor is indeed only in the ECM, the accumulation of the peptide could be due to enhanced retainment of the peptide in the tumor matrix. Please address this by explaining the tumor targeting by the CSG peptide.

ANSWER: Solid tumours including RIP1-Tag5 tumours contain a mixture of laminin-nidogen-1 complex that:

  • overlaps with angiogenic blood vessels (comparable to the basement membrane expressed in normal tissues), and
  • forms avascular network of ECM.

CSG and CSG-targeted TNFa (Yeow et al. EMBO Mol Med 2019) and IO-NPs (this manuscript, Figures 1, 5 and 6) accumulate mostly around the avascular ECM. We have extensively showed in Yeow et al. 2019 that CSG binding is highly specific for tumour ECM and not ECM in normal tissue. We have now revised the following paragraph to discuss the possible reason for CSG-selective homing to tumour ECM.

Previously, we have shown that the target for CSG-binding, the laminin-nidogen-1 complex, is an extensive network formed throughout the RIP1-Tag5 tumours and overlaps with fibrillar collagens including collagen-1 [49]. RIP1-Tag5 tumours express two- to three-fold higher laminin-nidogen-1 complex than the normal exocrine pancreas. Importantly, unlike normal tissue ECM that is expressed exclusively as a thin basement membrane supporting vessels and epithelia, in tumours, laminin and nidogen-1 in are also aberrantly expressed as avascular ECM [49]. These findings raise the possibility that structural differences in tumour ECM, absent in normal ECM, expose an otherwise hidden epitope for CSG binding. CSG binding is highly specific for tumour ECM and not ECM in normal tissue [49]. Analysis of IO-NP distribution in RIP1-Tag5 tumours shows CSG-IO-NP are dispersed within tumour ECM, well away from the nearest angiogenic vessels, with the overall intratumoral amount of CSG-IO-NP at least double that of CREKA-IO-NP. Here, we show the abundance of ECM in RIP1-Tag5 tumours is consistent in small and large tumours resulting in reliable intratumoral accumulation of CSG-IO-NP, unlike CREKA-IO-NP, which are dependent on tumour size and blood vessel content.

In addition, we also revised the last 3 paragraphs to improve the discussion.

Reviewer 2 Report

The current manuscript provides an interesting and detailed account of targeted delivery of iron oxide nanoparticles to tumour-specific ECM. The experiments are well conducted and the results are well reported. I recommend minor comments for the manuscript as follows:

1. I suggest adding data related to the magnetic properties of the developed system to know the effect of nonapeptide conjugation on these properties.

2. Also, the same applies to the nonapeptide properties and morphology after conjugation to the iron oxide NPs. Circular dichroism may be applied for the same.

Author Response

Dear Reviewer,

Thank you very much for your comments and support for our manuscript.

Whilst we have not assessed the changes to secondary structure of our peptides when they are in free in solution versus being tethered to the nanoparticle surface, we have used similar approach of peptide conjugation to nanoparticles as reported for other tumour homing peptides with less than 10 amino acids (examples cited in this paper: Agemy et al. Blood 2010; Agemy et al. PNAS 2011; Sugahara et al. Cancer Cell 2009; Park et al. 2009).

Importantly, our data show conjugated NPs retain the intratumoural binding specificity, comparable to the free peptides.  

Reviewer 3 Report

This paper entitled “Enhanced detection of desmoplasia by targeted delivery of iron oxide nanoparticles to tumour-specific extracellular matrix” exploits the used of ECM targeted iron Nps as pancreatic cancer imaging tool. This is more than interesting work, suitable for publication in Pharmaceutics. In general, the paper is well structured, well-written and clear. Here there are some comments/ suggestions that can be considered:

-INTRODUCTION: This section is more than clear and well structured. I just suggest including more information about ECM (composition, specially to justify the use of the nonapeptide that binds to entactin glycoprotein) and why the desmoplastic tumors are difficult to treat diseases.

-METHODS: I suggest adding more details in section 2.2 and 2.3. I mean: tumor generation, perfusion (transchardial?), number of mice, fixation I guess that PFA4%, tumor sample histological examination by microscopy…

Section 2.4: If I understood properly, iron Nps were coated with dextran and then encapsulated in the lipids with and without the peptide? This is correct? And was not dextran bound to that lipids, right?

Statistical analysis is missing in methods.

I am curious. Why did the authors select 1 h after peptide administration? Did they test lower times (e.g. 30min) and why 4 hours with coated Nps?. Were these time points based on previous studies?.

Author Response

Dear Reviewer,

Thank you very much for your feedback and support for our manuscript. We include below our revised manuscript and response to your comments/suggestions:

-INTRODUCTION: This section is more than clear and well structured. I just suggest including more information about ECM (composition, specially to justify the use of the nonapeptide that binds to entactin glycoprotein) and why the desmoplastic tumors are difficult to treat diseases.

ANSWER: We have revised the discussion section to include more information on the tumour ECM that makes them unique for CSG targeting (lines 411-425). We have added an explanation in the introduction on why desmoplastic tumours are difficult to treat (line 92-98).

-METHODS: I suggest adding more details in section 2.2 and 2.3. I mean: tumor generation, perfusion (transchardial?), number of mice, fixation I guess that PFA4%, tumor sample histological examination by microscope.

ANSWER: Sections 2.2 and 2.3 have been revised to include the required information. Specifically:

Tumour generation: Mice were used at 28-30 weeks of age as they typically developed multiple advanced tumour nodules. Perfusion: transcardial perfusion. Number of mice: A minimum of 3 mice/ group. Fixation: for peptide binding, tissues were kept as fresh frozen unfixed and for nanoparticle homing, tissues were fixed with 2% formalin (section 2.7). Histology: Section 2.9.

Section 2.4: If I understood properly, iron Nps were coated with dextran and then encapsulated in the lipids with and without the peptide? This is correct? And was not dextran bound to that lipids, right?

ANSWER: In this study, IO-NP were first coated with dextran and then the dextran-coated IO-NPs were encapsulated within the lipids. There is no covalent linkage between the dextran and the lipids. We add minor correction to section 2.4, which is complementary to the schematic diagram of the nanoparticles as shown in Figure 3.

Statistical analysis is missing in methods.

ANSWER: Our revised manuscript includes Section 2.10: statistical analysis.

I am curious. Why did the authors select 1 h after peptide administration? Did they test lower times (e.g. 30min) and why 4 hours with coated Nps?. Were these time points based on previous studies?

ANSWER: Yes, the circulation time used in this study is based on our previous studies (cited in this paper: Yeow et al. EMBO Mol Med. 2019; Hamzah et al. J Immunol 2009; Johansson et al. PNAS 2012; Johansson-Percival et al. Cell Report 2015).